# Repeated Menthol Mouth Swilling Affects Neither Strength nor Power Performance

**DOI:** 10.3390/sports8060090

**Published:** 2020-06-17

**Authors:** Russ Best, Dani Temm, Holly Hucker, Kerin McDonald

**Affiliations:** 1Centre for Sport Science and Human Performance, Wintec, Hamilton 3200, New Zealand; danitemm@gmail.com (D.T.); holly.hucker@gmail.com (H.H.); Kerin.McDonald@wintec.ac.nz (K.M.); 2School of Health and Life Sciences, Teesside University, Middlesbrough TS1 3BX, UK

**Keywords:** menthol, strength, power output, repeated sprint, vertical jump

## Abstract

This study aimed to assess the effects of repeated menthol mouth swilling upon strength and power performance. Nineteen (10 male) participants completed familiarisation and experimental trials of repeated menthol mouth swilling (0.1% concentration) or control (no swill) in a randomised crossover design. Participants performed an isometric mid-thigh pull (IMTP; peak and mean force; N), vertical jump (peak; cm) and six second sprint (peak and mean power; W) under each condition. Participants completed three efforts per exercise task interspersed with three-minute recoveries. Mean best values were analysed via a two-way mixed repeated measures ANOVA, and differences reported as effect sizes ± 95% confidence intervals, with accompanying descriptors and *p* values. Differences in peak IMTP values were unclear between familiarisation and experimental trials, and between menthol and control conditions. Mean IMTP force differed between familiarisation and control (0.51; −0.15 to 1.14; *p* = 0.001) and familiarisation and menthol conditions (0.50; −0.15 to 1.14; *p* = 0.002) by a small degree, but were unclear between control and menthol conditions. Unclear differences were also noted on vertical jump performance compared to familiarisation and between experimental conditions, with repeated six second peak and average power performance also showing unclear effects across all comparisons. We conclude that repeated menthol mouth swilling does not improve strength or power performance.

## 1. Introduction

Menthol is a botanical compound extracted from plants of the *mentha* genus (e.g., peppermint, corn mint), commonly used to impart a minty flavour and sensations of oral cooling and freshness [1]. Although research conducted upon the beneficial effects of chronic peppermint oil ingestion upon aerobic parameters [2] has since been refuted [3], menthol and peppermint extracts have been linked to feelings of alertness and improved decision making [4,5]. Menthol mouth swilling is also increasingly researched in endurance sports performance [6,7], where effects include increases in ventilation and improvements in thermal comfort and sensation [6,7]. These benefits have not been shown to manifest in improved repeated sprint performance [8] however, suggesting divergent responses across exercise intensities and durations, and consequently sporting modalities. Thus, the assessment of effects that may prove beneficial during explosive activities in sports such as weightlifting, BMX racing, rugby or field events present a natural progression to this line of enquiry.

Improvements in strength and power performance have been associated with other aromatic compounds. For instance, low concentration ammonia salts (smelling salts) are commonly used by weightlifters to enhance performance [9], despite an equivalence of findings within the literature [9,10,11]. Ingestion of capsaicin, the botanical compound responsible for chilli’s heat, has also been shown to improve resistance exercise performance [12].

Conversely, oral or nasal menthol administration stimulates transient receptor melastatin 8 proteins (TRPM8; [6,7]), which detect environmental cold. The ergogenic effects of menthol in hot environmental conditions are attributed to improved thermal perception or alterations in respiratory mechanics [6,13]. Mechanistically, whilst menthol application releases calcium from endoplasmic reticula and Golgi cells and increases intracellular calcium concentrations [14], it may concomitantly inhibit sodium and calcium channel activity [14,15]; thus, is unlikely to facilitate strength or power performance.

There has been only one investigation into menthol administration and strength and power performance to date [15]. Whilst significant improvements in isometric grip strength (36.1%; *p* < 0.05), vertical jump (7.0%; *p* < 0.01), standing long jump (6.0%; *p* < 0.05) and respiratory parameters were observed five minutes post-supplementation [15], the conclusions of this research need to be considered cautiously given the acute experimental time frame (measures taken at five and 60 min post-supplementation) and lack of a crossover [15]. Results may be considered as either Hawthorne or placebo effects, and it remains uncertain as to whether this effect is repeatable.

Given the established use of aromatic compounds in resistance training, especially prior to maximal efforts [9,16], and the beneficial effects seen with the ingestion of capsaicin prior to strength and short-duration endurance performance [12,17], if menthol were to improve strength or power performance, it would likely be readily accepted by recreational or resistance trained athletes. The presence of ergogenic effects in this instance are unlikely, given that menthol antagonist capsaicin has been shown to improve strength and short-duration aerobic performance [12,17], with an accompanying plausible mechanistic explanation [18].

Due to only a single study [15] investigating the application of oral menthol and strength and power performance using a parallel groups design to date, the purpose of this investigation was to assess the effects of repeated menthol mouth swilling on strength and power performance in recreationally trained athletes, employing a crossover design and incorporating evidence of familiarisation. The absence of a plausible ergogenic physiological mechanism, and previous work showing improvements in strength performance following supplementation with the menthol antagonist capsaicin, suggests this topic warrants further exploration, in lieu of a hypothesis.

## 2. Materials and Methods

This study employed a randomised crossover design with respect to solution administration; however, performance testing order was kept consistent for all participants to minimise unexplained variance due to possible inhibition or potentiation effects [19,20]. Participants were afforded a familiarisation session for all performance tests, with subsequent testing sessions (Control or Menthol) conducted in a randomised counterbalanced order. Protocols used during the familiarisation session were identical to testing sessions but allowed participant to establish equipment positioning and gain coaching on each test. Prior to commencing all sessions, participants were asked to refrain from alcohol and/or caffeine for a minimum of 12 h, and heavy exercise that may prevent them from producing a ‘best effort’ for a minimum of 24 h. Testing sessions were 60-min in duration and included a 10-min self-selected warm up (Activities included, cycling, joint mobilisation movements, dynamic stretching, body weight squats, submaximal countermovement jumps and cycling sprint efforts), followed by three maximal mid-thigh pulls, countermovement vertical jumps and 6-s cycle ergometer Wingate efforts. Each effort and each exercise type were separated by a 3-min rest period (as per Figure 1). Furthermore, each testing session was separated by two to seven days to allow sufficient wash-out for treatment and any associated delayed onset muscle soreness, similar to other experiments investigating similar interventions upon repeated sprint performance [21]. All procedures were conducted in a climate-controlled laboratory under thermoneutral conditions (22 ± 1 °C) at midday (12 pm–1 pm) to minimise the effects of circadian variability upon performance. Thermoneutral (<28 °C) conditions were chosen to increase the potential transferability of findings to a wider range of sporting environments. Participants provided written informed consent prior to commencing their familiarisation trial. Ethical approval was obtained from the Waikato Institute of Technology Human Research Ethics Group (Approval reference: WTFE1824062018) and the trial was carried out in accordance with the Health Research Council New Zealand ethical guidelines.

### 2.1. Participants

Prior to study commencement, participants’ basic anthropometric data were collected (height, cm; weight, kg). Recreational training status was defined as having undertaken physical activity ≥ twice per week, for at least three months prior to study enrolment. If participants trained inconsistently, or had being doing so for less than three months, or not at all, then they were classed as untrained and excluded from participation; all participants met recreational training status criteria. Recreational athletes were chosen as they are deemed less likely to benefit from possible post-activation potentiation effects [19,20] and are less likely to suffer oxidative damage between sessions [22]. Participants included 10 males (24.6 ± 3.9 years, 180.0 ± 3.4 cm, 84.0 ± 8.5 kg) and nine females (20.2 ± 1.0 years, 168.3 ± 3.6 cm, 70.2 ± 6.9 kg). Females were asked about their menstrual status and recorded any contraceptive use as part of their pre-screening. All female participants reported contraceptive use.

### 2.2. Menthol Solution and Swilling

A 0.1% menthol solution was administered in 25 mL aliquots, prepared as per recommendations of Best et al. [1]: a 0.1% menthol solution was diluted from a 5% menthol and ethanol stock solution (Sigma-Aldrich Company Ltd. Dorset, England), which was made by dissolving menthol crystals ((-)-menthol; Sigma-Aldrich Company Ltd. Dorset, England) in ethanol, and then diluting to the experimental concentration using distilled water. Participants swilled the solution 60 s before attempting each exercise effort, for a duration of ~10 s, before expectorating. A placebo solution was not used as peppermint extracts from commercial manufacturers either contain menthol are under proprietary non-disclosure agreements so menthol concentration is unknown. Water may induce similar sensations of oral cooling as menthol [23]; hence, the control trial did not employ a placebo swill.

### 2.3. Isometric Mid-Thigh Pull Protocol

Average and peak isometric mid-thigh pull (IMTP) values were assessed using a customised testing rig consisting of two portable force plates (Pasco, Roseville, CA, USA) and perpendicular vertical uprights; uprights were pre-drilled at 2.5 cm increments to allow adjustment of a steel dowel to participants’ mid-thigh [24]. Dowel height was established during familiarisation testing and all testing was performed unshod to reduce variability brought about by the cushioning properties of participants’ footwear noted in pilot testing. The rig was connected to customised computer software (High Performance Sport, Auckland, New Zealand) that allowed for measurement of average and peak net force over a three-second pull duration (Newtons; N). Each athlete performed three IMTP trials, with instruction to pull upwards against the bar with maximal effort while pushing their feet into the force plates; trials were separated by three-minute recoveries. Athletes self-selected either an overhand grip or an alternate (one hand pronated, one supinated) grip at their pre-established dowel height, and adopted this grip throughout all trials. Participants commenced each trial with feet shoulder width apart and positioned centrally upon each force plate(s), their hands were placed on the bar in preparation of the testing effort. Once body position was established, participants were given a three second countdown, into a three second maximal effort pull. All athletes were given verbal encouragement throughout the test. Upon completion of the effort, participants remained on the force plates briefly until values returned to a relative zero. Reliability of IMTP assessments of peak force have previously been reported as being extremely reliable within and between sessions (ICC: 0.96–0.98; 90% C.I. 0.90 to 0.98; [25,26]).

### 2.4. Vertical Jump Protocol

Vertical jump height (cm) was measured using a Vertec (Sports Imports, Hilliard, OH, USA) with each participant afforded three jumps per testing session, interspersed with three minutes of passive rest between each effort to facilitate maximal effort. To determine standing reach height (cm), each participant stood side on with their dominant arm nearest the Vertec while keeping their feet on the ground and reaching up with an elevated shoulder to displace the highest vane possible. Preliminary steps or shuffling before take-off or during the eccentric phase were not permitted, with participants also required to take-off from two feet. To ensure proper form, participants were instructed to begin in a standing position with their arms not raised past shoulder height, then perform a countermovement jump establishing a self-selected depth. Participants were further instructed to displace the highest vane possible at the apex of their jump. The initial standing reach height was then subtracted from the maximal height reached to find the participants vertical jump height. Intersession reliability of the Vertec has previously been reported as ICC: 0.80 for females and ICC: 0.90 for males [27].

### 2.5. Repeated Six Second Peak Power Protocol

Repeated six second efforts were performed on a Wattbike™ cycle ergometer (Wattbike Pro; Wattbike; Nottingham, UK) and were interspersed with three minutes of passive recovery. Seat and handlebar position were at the discretion of each participant, with participants encouraged to select ergometer geometry that they felt was most conducive to their performance; this position was maintained across testing sessions. Participants performed three maximal efforts remaining seated throughout and feet secured into pedals. Resistance was applied via airbrake (range: 3–8) and magnetic resistance (fixed at 1) relative to participants’ bodyweight, as per the manufacturer’s six second peak power test protocol [28]. Each trial began with a five second countdown, and following a verbal command of ‘when you’re ready’ participants completed the test from a standing start, leading with their preferred leg at a self-selected crank angle (~45°). Maximum and average power outputs (W) over the six second effort were sampled at 100 Hz and recorded for each trial. Wattbikes™ are considered a valid alternative to a Monark ergometer (*r* = 0.95; R^2^ = 0.9; SEE = 61.4 W; *p* < 0.0001; [29]). Wattbikes are factory calibrated so detail as to calibration is proprietary, although to ensure consistency between participants a factory reset was performed between participants.

### 2.6. Statistical Analyses

Participants mean best values from each activity, in each condition were used for analyses. Mean best scores were calculated as a mean of participants’ best two efforts per test, per condition [24,30]. Data were analysed using SPSS (v24; IBM, New York, NY, USA); data were considered normally distributed as per the following criteria: Shapiro-Wilks test *p* > 0.05, skewness and kurtosis were within ± 1, if the mean and median were within 10% of each other, or if 2 × SD > mean [31,32,33]. A two-way mixed repeated measures ANOVA was performed to assess the effects and interaction of treatment and participant sex, with an a priori alpha level of 0.05. Partial eta squared (partial η^2^) is provided to quantify the magnitude of the effect at the ANOVA level. Effect sizes (Cohen’s *d*) were calculated using a customised spreadsheet, and are described as follows: 0–0.2 trivial; 0.2–0.6 small; 0.6–1.2 moderate; 1.2–2.0 large; >2.0 very large [34], for pairwise comparisons. If an effect had a 95% confidence interval (CI) that crossed zero and exceeded the threshold for a small effect, it was deemed unclear [35]. Data are reported as effect sizes ± 95% CI at the group level (treatment) and between sexes, with accompanying statistical significance. Figures depicting individual and group change scores were produced via a freely available data visualisation tool [36], with differing symbols used to denote each biological sex.

## 3. Results

Maulchly’s test of sphericity was non-significant for all performance measures; thus, sphericity was assumed (*p* = 0.508–0.948). The ANOVA showed that there was a significant within participant effect of treatment across all variables, presenting with medium to large effects as per partial η^2^ values. Specifically, IMTP peak force *F* (2, 34) = 9.597, *p* < 0.001, partial η^2^ = 0.361; IMTP mean force *F* (2, 34), = 12.353, *p* < 0.001, partial η^2^ = 0.421; Vertical jump height *F* (2, 34) = 4.919, *p* = 0.013, partial η^2^ = 0.224; Repeated sprint peak power *F* (2, 34) = 6.157, *p* = 0.005, partial η^2^ = 0.266 and repeated sprint mean force *F* (2, 34) = 3.767, *p* = 0.033, partial η^2^ = 0.181. Similarly, the between subject effect of participant sex was also significant across all variables. Specifically, IMTP peak force *F* (1, 17) = 31.420, *p* < 0.001, partial η^2^ = 0.649; IMTP mean force *F* (1, 17) = 33.859, *p* < 0.001, partial η^2^ = 0.666; Vertical jump height *F* (1, 17) = 9.432, *p* = 0.007, partial η^2^ = 0.357; Repeated sprint peak power *F* (1, 17) = 25.238, *p* < 0.001, partial η^2^ = 0.598 and repeated sprint mean force *F* (1, 17) = 27.914, *p* < 0.001, partial η^2^ = 0.622. There was no significant interaction effect (treatment × sex) across any performance measure (all *p* > 0.05). The following sections detail comparisons between familiarisation and experimental trials, between experimental conditions and differences between participant sexes.

### 3.1. Familiarisation and Experimental Trials

Small differences in peak force production between familiarisation and experimental trials were noted for both control (ES: 0.43; 95% CI: −0.22 to 1.06; *p* = 0.001) and menthol swilling (0.36; −0.28 to 1.00; *p* = 0.011); however, due to confidence limits overlapping the threshold for a small effect, these effects are deemed unclear at the group level. With respect to average force production over the duration of the mid-thigh isometric pull, small differences were again seen between familiarisation and control (0.51; −0.15 to 1.14; *p* = 0.001) and familiarisation and menthol conditions (0.50; −0.15 to 1.14; *p* = 0.002) at the group level.

Differences in jump height between familiarisation and experimental trials were small and unclear (control: 0.22; −0.42 to 0.85; *p =* 0.075 and menthol: 0.28; −0.37 to 0.91; *p =* 0.051) due to confidence intervals overlapping small effect boundaries in both directions.

Peak power output attained during the repeated six second peak power protocol was trivially different between familiarisation and experimental conditions (control: 0.15; −0.49 to 0.78; *p =* 0.008 and menthol: 0.09; −0.55 to 0.73; *p =* 0.120) and between. The same trend was apparent when average power output was examined between familiarisation and experimental conditions (control: 0.10; −0.53 to 0.74; *p =* 0.087 and menthol: 0.10; −0.53 to 0.74; *p =* 0.099).

### 3.2. Experimental Trials

Unclear trivial differences were seen between control and menthol conditions (0.03; −0.61 to 0.67; *p* = 1.000) for IMTP peak force, with control and menthol conditions also differing trivially, but deemed unclear due to confidence interval breadth (0.01; −0.62 to 0.65; *p* = 1.000) for IMTP mean force production. Similarly, differences in vertical jump performance between control and menthol trials were trivial and unclear based on the confidence interval of the effect (0.05; −0.59 to 0.69; *p =* 1.000). Peak power output attained during the repeated six second peak power protocol was trivially different between experimental conditions (0.06; −0.58 to 0.70; *p* = 0.486), but broad confidence intervals rendered the effects unclear, likewise with respect to average power sustained over the six second trial(s) (0.00; −0.63 to 0.64; *p* = 1.000). Group responses are depicted in Figure 2, with individual responses shown in Figure 3.

### 3.3. Differences between and within Participant Sexes

Please note due to the uniformity of responses for between sex comparisons, effect sizes (ES) are presented as a range across comparisons, with an accompanying description of magnitude. Figure 3 highlights individual participant responses, including participant sex.

Males produced a mean of 752.30 N (95% CI: 469.14 N to 1035.46 N; *p* < 0.001) more force during the IMTP than female participants (ES: 2.23 to 2.56; very large) and similarly sustained a greater force throughout the trials (IMTP average: 411.32 N; 262.18 N to 560.46 N; *p* < 0.001; ES: 2.12 to 2.56; very large). Vertical jump performance was also significantly higher in males than in females (8.99 cm; 2.81 to 15.87 cm; *p* = 0.007; ES: 1.26 to 1.47; large). Likewise, males demonstrated greater peak (448.44; 260.11 to 636.77; *p* < 0.001; ES: 2.10 to 2.27; very large) and mean (401.48 W; 241.16 W to 561.80 W; *p* < 0.001; ES: 2.29 to 2.41; very large) power outputs in the repeated six second peak power protocol.

In male athletes, all effects between conditions were rendered unclear due to confidence intervals overlapping zero, with the exception of IMTP mean force, which was moderately different between familiarisation and menthol conditions (0.82; −0.13 to 1.69; *p* = 0.08). Similarly, female participants showed moderate improvements in IMTP mean force production between familiarisation and control (0.94; −0.08 to 1.86; *p* = 0.06) and familiarisation and menthol trials (0.86; −0.14 to 1.78; *p* = 0.09). Female participants also elicited a moderately greater IMTP peak power in their control trial compared to their familiarisation (0.79; −0.20 to 1.71; *p =* 0.11).

## 4. Discussion

This investigation assessed the effects of repeated menthol mouth swilling on strength and power performance in recreationally trained athletes. Menthol mouth swilling only trivially affected performance, relative to control trials, across all performance tests undertaken, in normothermic conditions.

Our data refute previous evidence that acute exposure to menthol improves lower-limb power performance [15]; we have demonstrated this in a pre-post crossover design, allowing sufficient time for treatment washout between trials, in male and female recreationally trained athletes. This approach would typically be considered stronger than that of Meamarbarshi, who employed a parallel groups trial that lasted ~60 min [15], sampling data at five- and 60-min post menthol exposure. In a crossover design, individual as well as between condition change scores can be calculated, identifying any participants who respond atypically to an intervention or establishing a ‘normal’ response, whereas a lack of intra-individual controls and change scores limits the certainty with which conclusions can be applied beyond a group and testing occurrence.

A short time experimental frame is also problematic, as different mechanisms may be attributable to performance enhancements seen at each time point. Indeed, Meamarbashi [15] states that mechanisms responsible for the relatively large improvements in lower (6.4–7.0%; raw mean change: 2.7–5.1 cm) and upper limb power (17.5–36.4%; raw mean change: 6.7–10.4 kg) seen in their work are unknown. These changes may be explained by an initial learning effect, followed by typical variation within the test(s), and the time to peak plasma menthol concentration of 30–120 min [37], (half-life of ingested menthol of ~40–60 min [37]). Thus, depending upon an individual’s metabolic response to menthol, they are likely to be at or near peak plasma menthol concentration when the final testing bout took place at 60 min post menthol ingestion, as well as having had three attempts at the exercise(s)—the effect is thus a combination of metabolic timing and/or familiarisation, as these responses cannot be differentiated under such a design. These effects may also be a result of increased stimulation of receptors along the alimentary canal, as per other ergogenic tastants that differ in ergogenic effects when swilled or ingested [38].

Enhancements of previously reported magnitudes appear even more questionable given no changes in electromyography or tissue oxygen saturation have been demonstrated following repeated oral menthol exposure [39]. Whilst menthol may cause a net calcium efflux, menthol may subsequently inhibit the reuptake of sodium and calcium by the cell, as well as limiting associated enzyme activity [13,14]. Practically, this may partially explain both Meamarbashi [15] and our findings, whereby a single dose of menthol may prove effective, but repeated application may act to inhibit contractile or associated enzymatic activity. It should be noted that this mechanism has also been proffered as an explanation for menthol’s analgesic effect [40], with alterations in regional cutaneous blood flow also observed [39], thus warranting investigation into repeated topical application of menthol during strength and power performance, beyond the work already conducted in facilitating recovery [41,42,43].

Some participants complained of acute flavour fatigue towards the end of the menthol trial, possibly due to an overstimulation of oral cold receptors, and a resultant unpleasant mouth feel; this itself may have detracted from any potential ergogenic effects. Similar adverse reactions have been reported following capsaicin ingestion prior to repeated sprint exercise [44], but were not reported by Gibson et al. [8] following a capsaicin or menthol swill during a repeated sprint exercise, albeit no performance enhancement was found following either menthol or capsaicin swilling [8]. Capsaicin may act as an irritant when applied in the oral cavity [45], whereas menthol is renowned for its cooling and calmative properties [23,46], minimising the likelihood of adverse reactions if applied in appropriate concentrations [1] and possibly ensuring the strategy is adopted by athletes or recreational exercisers. Interestingly, power-lifting athletes report using smelling salts most frequently during dead-lift performance and only for two to three efforts during competition [9]; the dead-lift is typically performed last during competition, suggesting athletes prefer to use aromatic containing strategies when they are likely to be most fatigued. It may be that athletes who enjoyed the sensation and taste imparted by menthol in the present investigation may benefit from swilling under self-selected and/or pre-fatigued circumstances. Indeed, a single menthol swill has been shown to extend time to exhaustion in endurance performance following a pre-fatiguing period [47].

Despite finding no meaningful differences between experimental conditions across all metrics assessed, this trial emphasises the importance of familiarisation testing, especially in trials which incorporate an intervention, or may be susceptible to placebo, nocebo or Hawthorne effects [48,49]. Familiarisation tests serve to minimise the variability within participants’ performance, which is evident in the present investigation as standardised mean differences were greater between familiarisation and experimental trials (Small), than between experimental conditions (Trivial). However, familiarisation sessions should be optimised for trial activities. Due to the small-moderate differences in performance between familiarisation and control and or menthol trials in some tests, this may not have been the case in the present investigation. Intra-individual variability, and thus, the likelihood of committing a type one error was further minimised by assessing trained participants at approximately the same time of day, allowing for sufficient recovery between conditions, in accordance with previous research [21].

The present investigation has several limitations: Firstly, whilst familiarisation and control treatments were included, participants were not blinded to the purpose of the study, either qualitatively or through the use of a placebo swill, as there is no appropriate taste matched placebo treatment for menthol. Peppermint extracts must by their nature contain menthol, and manufacturers were reluctant to disclose menthol percentage due to patents and perceived risk to intellectual property. The authors acknowledge that whilst water may not be an ideal control swill, matching the solution to environmental or oral temperature [50] would aid in elucidating experimental effects. An alternative may have been to blind participants to the purpose of the study, and use a non-taste-matched placebo. Secondly, although we recorded oral contraception use in female participants, the nature of the contraception used was not documented. This is an important consideration and potentially useful covariate for future investigations, as mono- or multiphasic contraceptives have different effects upon participants’ circulating hormone levels [51]; thus, performance may be affected differently at differing times of the cycle, although to date, menstrual phase, associated symptoms and/or oral contraceptive use have shown equivocal effects with respect to power output in trained females [52,53,54,55].

## 5. Conclusions

In conclusion, repeated menthol mouth swilling affects neither strength nor power performance in recreationally trained males or females. These results refute previous work in this area [15], whilst highlighting the importance of replication and documenting familiarisation trials in research involving exercise assessment and nutritional interventions. However, whilst repeated exposure to odorant or flavour interventions (e.g., menthol) may not improve strength or power performance, some athletes may feel that acute use of these strategies e.g., later in competition, is of benefit—this remains to be empirically assessed with respect to menthol, but applied [15] and mechanistic [13,14] evidence suggests ergogenic effects are plausible; these may be partially mediated by placebo effects [38].

## Figures and Tables

**Figure 1 sports-08-00090-f001:**
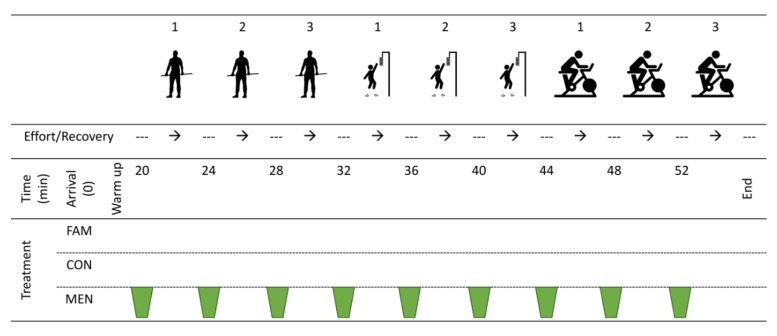
Experimental schematic, detailing testing procedures, exercise and recovery intervals and swilling time points.

**Figure 2 sports-08-00090-f002:**
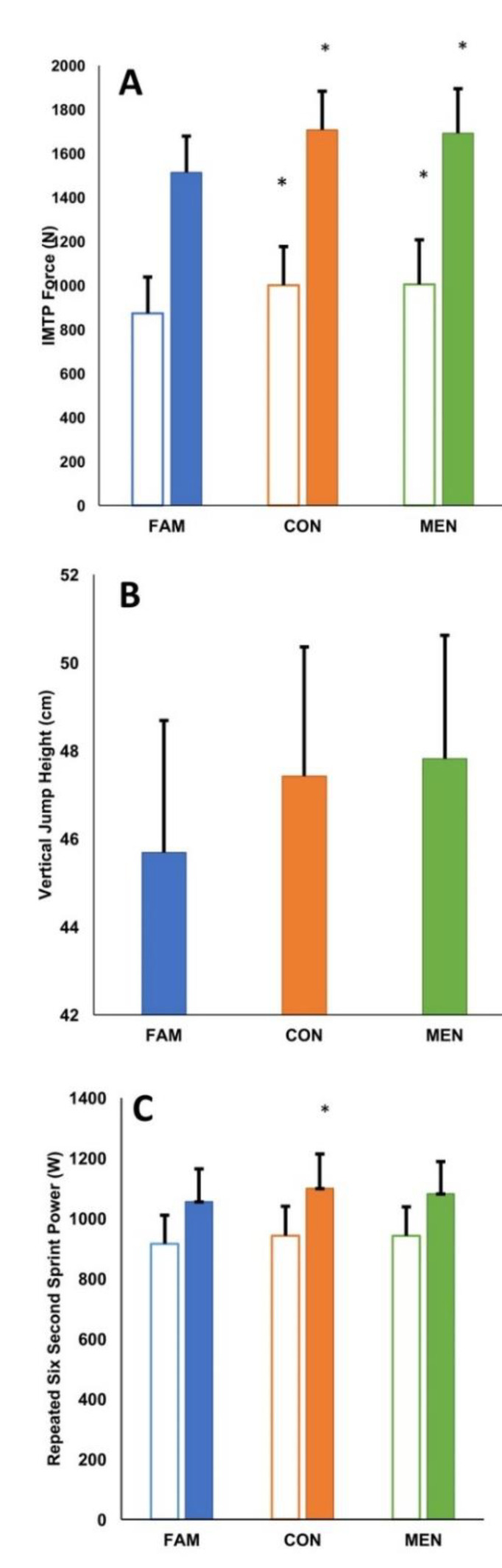
Group mean responses for peak force produced and average force sustained (**Panel A**) during an isometric mid-thigh pull, vertical jump height (**Panel B**) and peak and average power output during a repeated six second sprint protocol (**Panel C**). Clear bars denote average values and asterisks (*) denote a statistically significant difference to familiarisation of *p* < 0.05.

**Figure 3 sports-08-00090-f003:**
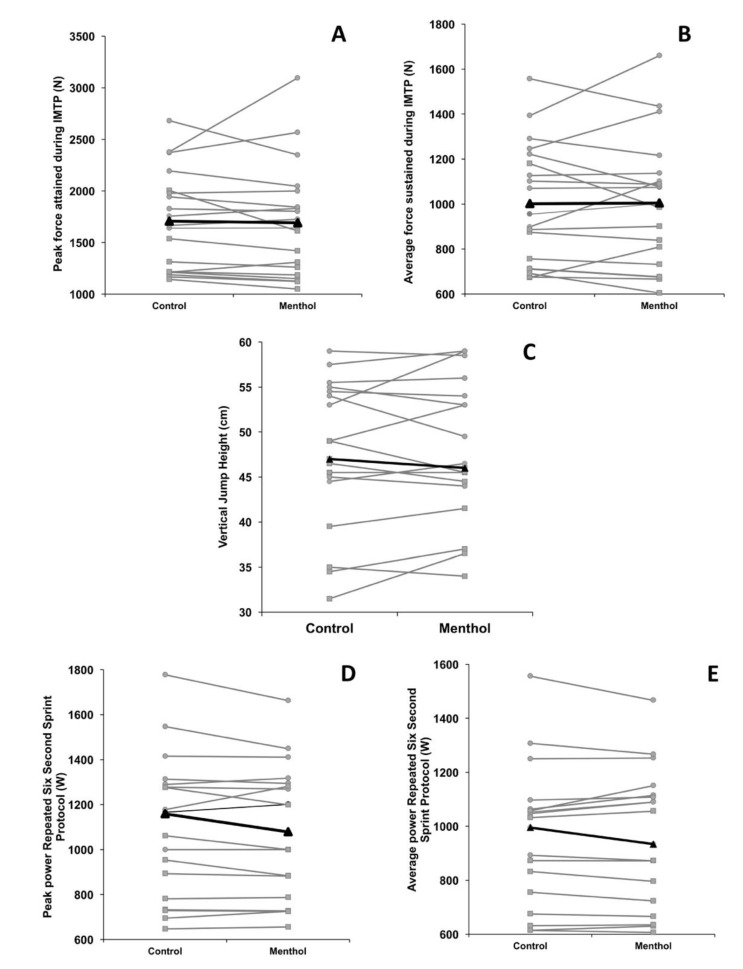
Individual response(s) for experimental trials for isometric mid-thigh pull (**Panels A,B**), vertical jump (**Panel C**) and repeated six second peak power protocol (**Panels D,E**). Mean response is denoted by the thick black line (triangle markers), female and male participants are represented by square (■) and circle markers (●), respectively.

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
