# Peer review of "Repeated Menthol Mouth Swilling Affects Neither Strength nor Power Performance"

_sports, 2020, doi:10.3390/sports8060090_

Round 1
Reviewer 1 Report
The reviewer would like to thank the authors for the opportunity to review their work. Given the dearth of literature examining the influence of menthol mouth swilling this paper is welcomed and a useful additon to the field. The paper proposes to examine the effects of repeated menthol mouth swilling on strength and power performance in recreationally trained participants. Ten male and nine female participants completed three trials including familiarisation, menthol mouth swilling and control between completion of isometric mid thing pull, vertical jump and 6-s sprints. Findings indicate repeated menthol mouth swilling affects neither strength nor power performance in recreationally trained males or females.
Comments:
Line 61 - 97: Please reconsider the structure of this paragraph as understanding the main point is difficult. Is it implying that a placebo effect is the most likely explanation if menthol improved strength performance in recreational athletes? Is this based on the fact that capsacin improves strength performance and therefore as an antagonist to menthol, menthol couldn't achieve the same effect? Please clarify.
General comment about methods:
Given the difficulty with establising a placebo solution for methol studies can the authors explain why the participants in this study were not blinded as to the intentions of the study as this would have produced a more robust data set. Further, while I accept that water may induce similar sensations of oral cooling as menthol is a no swill control an appropriate substitute? Surely having one trial where swilling occurs and other trials with no swilling creates a situation where any potential placebo effect is reinforced without being able to truly disentagle whether menthol is having a real effect on performance. If an effect of menthol swilling was found in this study would it have been possible to conclude that this was soley due to menthol given the current design? Perhaps water at a controlled temperature close to body temperature (e.g 36°C as in the work of Eccles which has been shown not to stimulate cold receptors in the mouth) may have been a better substitute as the swilling process would then have been consistent across trials? Please justify your thinking here.
Line 80: Please elaborate on the familiarisation trial and what the familiarisation entailed for each of the performance tests. It appears that the familiarisation was identical to the main trials but there is existing evidence e.g. with respect to repeated sprinting, of what is required during familiarisation to minimise a learning effect. I imagine similar exists for IMTP and vertical jump and therefore could the authors explain their thinking and approach to the familiarisation process.
Line 132: Please report the reliability of the Vertec system
Line 136: Check spelling of 'vein'. I believe this should be 'vane'.
General comment about results:
Line 161-162 of the statistical analyses section identifies that a one way ANOVA was performed between familiarisation and each experimental condition and between control and menthol trials. Given that the analyses could likely have been performed with a two way ANOVA instead could the authors explain their reasoning for this approach? Further to this, could the authors report the findings of the ANOVA in the results section prose (F value and degree of freedom).
Line 164-165 of the statistical analyses section states that analyses were performed at the group level and then repeated for male and female participants. However, there is no reference to the results for male and female participants in the results section. Perhaps this is because there was no difference / effect, however, acknowledgement of this should be made by inserting a sentence at the appropriate point in each section of the results.
Discussion:
Line 270 - 278: I agree with the comment here regarding the importance of a familiarisation session. However, it is equally important that the familiarisation session be optimal such that the potential for learning is minimised. From reading I do not believe that familiarisation was optimised in the current study as there is no indication that the familiarisation session was based on existing evidence and there is evidence of a learning effect from familiarisation to main experimental trials. Therefore, the authors should acknowledge this in this paragraph.
Author Response
Response to Reviewer 1 -
The reviewer would like to thank the authors for the opportunity to review their work. Given the dearth of literature examining the influence of menthol mouth swilling this paper is welcomed and a useful additon to the field. The paper proposes to examine the effects of repeated menthol mouth swilling on strength and power performance in recreationally trained participants. Ten male and nine female participants completed three trials including familiarisation, menthol mouth swilling and control between completion of isometric mid thing pull, vertical jump and 6-s sprints. Findings indicate repeated menthol mouth swilling affects neither strength nor power performance in recreationally trained males or females.
We thank the reviewer for their appraisal of our work, and acknowledging that the paper adds to this research area, given the relative paucity of literature.
Comments:
Line 61 - 97: Please reconsider the structure of this paragraph as understanding the main point is difficult. Is it implying that a placebo effect is the most likely explanation if menthol improved strength performance in recreational athletes? Is this based on the fact that capsacin improves strength performance and therefore as an antagonist to menthol, menthol couldn't achieve the same effect? Please clarify.
General comment about methods:
Given the difficulty with establising a placebo solution for menthol studies can the authors explain why the participants in this study were not blinded as to the intentions of the study as this would have produced a more robust data set. Further, while I accept that water may induce similar sensations of oral cooling as menthol is a no swill control an appropriate substitute? Surely having one trial where swilling occurs and other trials with no swilling creates a situation where any potential placebo effect is reinforced without being able to truly disentagle whether menthol is having a real effect on performance. If an effect of menthol swilling was found in this study would it have been possible to conclude that this was soley due to menthol given the current design? Perhaps water at a controlled temperature close to body temperature (e.g 36°C as in the work of Eccles which has been shown not to stimulate cold receptors in the mouth) may have been a better substitute as the swilling process would then have been consistent across trials? Please justify your thinking here.
The rationale for a no swill comparison was to mimic that of ammonia use by strength athletes, which presents a pungent stimulus, implemented at a specific time. Qualitative reports of ammonia use in strength athletes suggest that this strategy is either implemented or not. To the authors knowledge no alternatives are employed.
We acknowledge the lack of control swill or appropriately blinded placebo is a limitation, however since alterations in legislation brought about due to a death associated with caffeine ingestion in 2018, obtaining institutional ethical approval for studies that involve ingestion of novel products has become increasingly difficult. The ethical committee felt that deception was inappropriate and would not approve the design under those conditions. Again, we acknowledge that a placebo swill would have strengthened the design and have included this as a limitation. We have also recommended environmental or bodily temperature matched water as a possible, although not ideal control swill.
Line 80: Please elaborate on the familiarisation trial and what the familiarisation entailed for each of the performance tests. It appears that the familiarisation was identical to the main trials but there is existing evidence e.g. with respect to repeated sprinting, of what is required during familiarisation to minimise a learning effect. I imagine similar exists for IMTP and vertical jump and therefore could the authors explain their thinking and approach to the familiarisation process.
It is correct that the familiarisation protocols were identical to the main trials. However, participants were given coaching on each test and allowed time to establish their individualised set up on equipment. Sub maximal trials were also allowed prior to testing to ensure safety and correct positioning.
In regard to minimising the learning effect, the researchers are in agreeance that allowing for a more robust familiarisation protocol to take place would be advantageous and note that Hopker (2009) and colleagues suggest at least 3 familiarisation sessions for sprinting. However, participant availability to an increased number of sessions was not viable under our current circumstances or time frames. It should also be noted that a study by Pekunlu (2015) indicated that a single familarization session with an extensive number of short cycling sprints can reduce variability. While Nibali (2015) and colleagues found that vertical jumps can be performed without the need for familiarisation.
Hopker, J. G., Coleman, D. A., Wiles, J. D., & Galbraith, A. (2009). Familiarisation and reliability of sprint test indices during laboratory and field assessment. Journal of sports science & medicine, 8(4), 528–532. states
Pekünlü, E. (2015). A large number of short sprints is required to avoid practice based improvement effect during the assessment of cycling peak power. Isokinetics & Exercise Science, 23(2), 133–142.
Nibali, M. L. , Tombleson, T. , Brady, P. H. & Wagner, P. (2015). Influence of Familiarization and Competitive Level on the Reliability of Countermovement Vertical Jump Kinetic and Kinematic Variables. Journal of Strength and Conditioning Research, 29(10), 2827–2835. doi: 10.1519/JSC.0000000000000964.)
Line 132: Please report the reliability of the Vertec system
Amended as requested
Line 136: Check spelling of 'vein'. I believe this should be 'vane'.
Amended as requested
General comment about results:
Line 161-162 of the statistical analyses section identifies that a one way ANOVA was performed between familiarisation and each experimental condition and between control and menthol trials. Given that the analyses could likely have been performed with a two way ANOVA instead could the authors explain their reasoning for this approach? Further to this, could the authors report the findings of the ANOVA in the results section prose (F value and degree of freedom).
We thank the reviewer for this comment, we have performed a two-way mixed repeated measures ANOVA, to assess the effects and interaction of participant sex and treatment. We have followed this with ES reporting for each sex, which addresses the reviewer’s following comment. Our apologies for neglecting to include this in the earlier draft.
Line 164-165 of the statistical analyses section states that analyses were performed at the group level and then repeated for male and female participants. However, there is no reference to the results for male and female participants in the results section. Perhaps this is because there was no difference / effect, however, acknowledgement of this should be made by inserting a sentence at the appropriate point in each section of the results.
Our apologies, this how now been amended as part of the restructured results section. Within and between sex comparisons are included as subsection 3.3 lines x – y.
Discussion:
Line 270 - 278: I agree with the comment here regarding the importance of a familiarisation session. However, it is equally important that the familiarisation session be optimal such that the potential for learning is minimised. From reading I do not believe that familiarisation was optimised in the current study as there is no indication that the familiarisation session was based on existing evidence and there is evidence of a learning effect from familiarisation to main experimental trials. Therefore, the authors should acknowledge this in this paragraph.
We agree with the reviewer’s appraisal and have included the following statement ‘However, familiarisation sessions should be optimised for trial activities. Due to the small differences in performance between familiarisation and control and or menthol trials in some tests, this may not have been the case in the present investigation.’
Reviewer 2 Report
Abstract
L12. It is not necessary to state the n for both sexes.
L13. Provide detail on frequency of ‘repeated’ swilling.
L16. Clearly state that peak/best or mean efforts were analysed?
L18 – 24. Is there scope to add data for these variables to the abstract?
Introduction
L31. Replace one instance of the double use of ‘upon’
L33. Suggest sentence break here. Suggest saving ‘this may prove beneficial during explosive activities in sports such as weightlifting BMX, rugby or field events.’ Until after the ergogenic effects of menthol have been outlined in L34-38.
L37. Provide a brief rationale as to why single power/strength bouts might be positively altered when repeated bouts are not.
L42-L43. Gibson et al., did not report ergogenic effects of Capsaicin in repeated sprint activity.
L42-L46. Are these sentences a necessary inclusion given Capsaicin was not tested in this study? The paragraphs commencing L48 and L54 follow on more logically from the paragraph that commences L39.
L46. Define excess.
L55. Are these significant improvements?
L71. The lack of a placebo swill is a major limitation of your study. Why was a comparison between Control and/or Placebo, and Menthol not investigated? It should clearly be stated that menthol swilling is compared to no swilling. I also suggest the language promoting the correctness of the experimental design of this study is toned down accordingly.
Materials and Methods
L86. Why was this environmental temperature selected?
L108. The rationale for not including a placebo is insufficient. The study was designed to determine the effects of swilling menthol diluted in water. As such there should be a water only control trial to isolate the independent effects of menthol. This could have been performed alongside a no-swilling trial. To increase the robustness of this work, a water swill vs no swill parallel study should be undertaken.
L158. Please define ‘mean best’
L163. Please clarify the statistical approach here. I suggest including an experimental schematic. To examine the effects of familiarisation, a comparison based upon test order, not trials should be conducted i.e. familiarisation vs visit 1 vs visit 2. Based upon comments regarding the results section, another recommendation would be to remove the comparisons with the familiarisation altogether. Confidence that the familiarisation had been sufficient could come from comparing visit 1 to visit 2 via paired sample t-tests. To examine the effects of menthol, paired sample t-tests should then be implemented.
Results
Based on the above, I suggest revising this section to first demonstrate an improvement from the familiarisation (section a – if deemed worthwhile for inclusion), and then section b includes the true comparison between experimental trials.
Consider demonstrating individual data as a scatterplot with a line of equality, this will aid with identifying if training status impacts the response arising from menthol. A panel figure will all performance variables should be included.
Discussion
L222. It should be stated that the trivial response favours control over menthol (Fig 3).
L240. Please clarify your comment here. To me, testing a participant when then are likely at peak plasma concentration is logical and the best way to determine whether ergogenic effects are likely.
L244. These are physiological responses which occur following true ingestion, is it likely that they will have occurred from rinsing only?
L264. Might assessing the effects of menthol during fatigue be a logical future study?
Conclusions
Reduce in length by 50%. Summarise the null finding, and provide simple take home message.
Author Response
Response to Reviewer 2 –
Abstract
L12. It is not necessary to state the n for both sexes.
Amended as requested.
L13. Provide detail on frequency of ‘repeated’ swilling.
L16. Clearly state that peak/best or mean efforts were analysed?
Amended as requested
L18 – 24. Is there scope to add data for these variables to the abstract?
Amended as requested
Introduction
L31. Replace one instance of the double use of ‘upon’
Amended as requested
L33. Suggest sentence break here. Suggest saving ‘this may prove beneficial during explosive activities in sports such as weightlifting BMX, rugby or field events.’ Until after the ergogenic effects of menthol have been outlined in L34-38.
Amended as suggested – this section L34 – 39 now reads as follows ‘Menthol mouth swilling is also increasingly researched in endurance sports performance [6,7], where effects include increases in ventilation and improvements in thermal comfort and sensation [6,7]. These benefits have not been shown to manifest in improved repeated sprint performance [8] however, suggesting divergent responses across exercise intensities and durations, and consequently sporting modalities. Thus, the assessment of effects that may prove beneficial during explosive activities in sports such as weightlifting BMX, rugby or field events, present a natural progression to this line of enquiry.’
L37. Provide a brief rationale as to why single power/strength bouts might be positively altered when repeated bouts are not.
L42-L43. Gibson et al., did not report ergogenic effects of Capsaicin in repeated sprint activity.
L42-L46. Are these sentences a necessary inclusion given Capsaicin was not tested in this study? The paragraphs commencing L48 and L54 follow on more logically from the paragraph that commences L39.
L46. Define excess.
The following sentences have been deleted and by doing so either amends or negates the above recommendations
‘Capsaicin’s ergogenic effects are attributed to an increase in calcium release from the sarcoplasmic reticulum, via stimulation of transient receptor potential vanilloid 1 proteins (TRPV1; [13]), which also act as environmental detectors of heat [14]. However, excess acute consumption of pungent capsaicin containing foods may lead to gastrointestinal distress, thus impairing performance.’
L55. Are these significant improvements?
Yes – data for handgrip and standing long jump were significant at p<0.05 and vertical jump at p<0.01. These values have been included in text for clarity.
L71. The lack of a placebo swill is a major limitation of your study. Why was a comparison between Control and/or Placebo, and Menthol not investigated? It should clearly be stated that menthol swilling is compared to no swilling. I also suggest the language promoting the correctness of the experimental design of this study is toned down accordingly.
Language amendments have been made as requested. The rationale for a no swill comparison was to mimic that of ammonia use by strength athletes, which present a pungent stimulus, implemented at a specific time. Qualitative reports of ammonia use in strength athletes suggest that this strategy is either implemented or not. To the authors knowledge no alternatives are employed.
We acknowledge the lack of control swill or appropriately blinded placebo is a limitation, however since alterations in legislation brought about due to a death associated with caffeine ingestion in 2018, obtaining institutional ethical approval for studies that involve ingestion of novel products has become increasingly difficult. The ethical committee felt that deception was inappropriate and would not approve the design under those conditions. Again, we acknowledge that a placebo swill would have strengthened the design and have included this as a limitation.
Materials and Methods
L86. Why was this environmental temperature selected?
Thermoneutral conditions were chosen to increase the potential transferability of findings to ecological settings. This statement has been included in the text.
L108. The rationale for not including a placebo is insufficient. The study was designed to determine the effects of swilling menthol diluted in water. As such there should be a water only control trial to isolate the independent effects of menthol. This could have been performed alongside a no-swilling trial. To increase the robustness of this work, a water swill vs no swill parallel study should be undertaken.
The researchers are in agreeance that further investigation and comparison utilising a water swill would add to understanding of the topic. The study in its current form however was only attempting to compare menthol swilling with no swill; similar to athletes who either utilise ammonia or not. At the present time the conducting of another trial/ parallel trial incorporating a water only trial is infeasible due to inability to access the laboratory and restrictions around social distancing.
L158. Please define ‘mean best’
We have amended the sentence that follows the initial occurrence of mean best to read as follows: ‘Mean best scores were calculated as a mean of participants’ best two efforts per test, per condition [23,28]’
L163. Please clarify the statistical approach here. I suggest including an experimental schematic. To examine the effects of familiarisation, a comparison based upon test order, not trials should be conducted i.e. familiarisation vs visit 1 vs visit 2. Based upon comments regarding the results section, another recommendation would be to remove the comparisons with the familiarisation altogether. Confidence that the familiarisation had been sufficient could come from comparing visit 1 to visit 2 via paired sample t-tests. To examine the effects of menthol, paired sample t-tests should then be implemented.
We thank the reviewer for this comment, we have performed a two-way mixed repeated measures ANOVA, to assess the effects and interaction of participant sex and treatment, with post-hoc comparisons also made. Furthermore, we have followed this with ES reporting for each sex. Our apologies for neglecting to include this in the earlier draft. We have included an experimental schematic as recommended, too.
Results
Based on the above, I suggest revising this section to first demonstrate an improvement from the familiarisation (section a – if deemed worthwhile for inclusion), and then section b includes the true comparison between experimental trials.
Consider demonstrating individual data as a scatterplot with a line of equality, this will aid with identifying if training status impacts the response arising from menthol. A panel figure will all performance variables should be included.
We thank the reviewer for this recommendation. The results have been restructured accordingly, but a scatterplot not included as we preferred to rearrange the figures to depict mean and individual responses separately. We hope this achieves the same goal the reviewer had in mind from the scatterplot.
Discussion
L222. It should be stated that the trivial response favours control over menthol (Fig 3).
This figure has now been amended, and the new figure 3 suggests there is a tendency for parity between menthol and control trials in three out of the five tests (as per mean lines/triangles).
L240. Please clarify your comment here. To me, testing a participant when then are likely at peak plasma concentration is logical and the best way to determine whether ergogenic effects are likely.
The following is now included at line 426/427 ‘the effect is thus a combination of metabolic timing and/or familiarisation, as these responses cannot be differentiated under such a design’
L244. These are physiological responses which occur following true ingestion, is it likely that they will have occurred from rinsing only?
The following has been added to distinguish between ingestion and swilling and highlight this as an area of future investigation: ‘These effects may also be a result of increased stimulation of receptors along the alimentary canal, as per other ergogenic tastants that differ in ergogenic effects when swilled or ingested [38].’
L264. Might assessing the effects of menthol during fatigue be a logical future study?
Yes, we agree with the reviewer’s comment and have included this as a future recommendation, citing previous literature when menthol mouth swilling was applied to similar effect in endurance performance.
Conclusions
Reduce in length by 50%. Summarise the null finding, and provide simple take home message.
Amended as requested
Reviewer 3 Report
The authors should be commended for exploring a novel topic and one that is applicable to some power-based sports. Overall, the methodology of this study has some major flaws – the methods are lacking detail, no deception was used or blinding to the purpose of the study, no placebo swill, minimal familiarization of the power based protocols. The authors acknowledge the limitations in the discussion, which is good, but it is alarming that the methods were not probably vetted prior to the administration of this study.
Please see the attached word document with specific comments.

Author Response
Response to Reviewer 3 –
Title:
-consider capitalizing “affects” and “neither” as well. Currently it is a bit eye catching. -Also, is it effects or “affects”? You state in the title it is ‘affects’ but in the first line of the abstract it is ‘effects’...
In the title affects is appropriate as the intervention is applied to affect performance, subsequently in the abstract effect is used to describe a resultant action from a causal factor i.e. the affect drives the effect, hence the difference between title and abstract.
Abstract:
-replace ‘average’ with mean...
Amended as requested
Introduction:
Ln 34 – comma between weightlifting and BMX
Ln 37-38 – you need to mention that the research in endurance performance was related to thermo-perception and thermal comfort in HOT or temperate conditions.
Both amended as requested
Please ensure the font for your reference citations is the same as the font for your document. Amended as requested.
Methods:
Please mention how the participants warmed-up prior to the test protocol.
Amended as requested, please find the following on Line 137-142 “Testing sessions were 60-minutes in duration and included a 10 minute self-selected warm up (Activities included, ergometric cycling, joint mobilisation movements, dynamic stretching, body weight squats, submaximal countermovement jumps and cycling sprint efforts)”
How long did each test session last? More detail is needed in the methods about the test day. Was nutrition controlled prior to the testing session? Hydration? You need to provide more detail on the methodology. How much time was between tests?
Some of the above factors are now detailed in the experimental schematic (Figure 1) and in text (see below comments). Nutrition was participants’ habitual diet, but participants were asked to refrain from alcohol and caffeine for a minimum of 12 hours prior to testing, as stated on line 135-137. Hydration status was not assessed.
Ln 107 – explain this more. The participant swilled the menthol solution between exercises – what was the mean time between exercises? How long was the entire testing session? I assume there was a total of 3 menthol swills?? It the participant swill prior to the first exercise test or only ‘between’. More detail is warranted in the methods.
An experimental schematic has been used to provide detail on these comments, and more clarification also included in text.
Lines 137-145 “Testing sessions were 60-minutes in duration and included a 10 minute self-selected warm up (Activities included, ergometric cycling, joint mobilisation movements, dynamic stretching, body weight squats, submaximal countermovement jumps and cycling sprint efforts), followed by three maximal midthigh pulls, countermovement vertical jumps and 6-second cycle ergometer Wingate efforts. Each effort and each exercise type were separated by a 4-minute rest period. Furthermore, each testing session was separated by two to seven days…”
Lines 170-171 “Participants swilled the solution 60-seconds before attempting each exercise effort, for a duration of ~10 seconds, before expectorating”
You mention in the methods that the washout time varied between 2 and 7 days. Explain why the ‘wash-out’ had so much variance? Considering the population tested are ‘recreationally trained’ with potentially no weight lifting or power-based training done before, is 2 days sufficient for adequate recovery to ensure a maximal effort on the subsequent test day?
Comment on the participants history of strength and power training. As the ‘recreational’ exclusion criteria of >2 days per week may have been entirely aerobic training. How did you ensure this was a homogenous sample of individuals with a history of power based training?
The variance between washout periods was due to participant availability. While some could commit to once a week other’s could not. Participants were for the most part students studying sport and exercise science and as such had regular exposer to resistance training through their tutorials as well as their own training. While tests were maximal in nature the duration of attempts were short with a total exercising time < 30 seconds. No participants expressed any fatigue or potential DOMs between sessions.
Why did you not have a control swill solution? How do you control for the placebo effect here in relation to performance? Deception or blinding could have been used informing the participant that you were assessing beverage preference prior to power based testing and a placebo could have been easily incorporated. This is a limitation to the study design.
The rationale for a no swill comparison was to mimic that of ammonia use by strength athletes, which present a pungent stimulus, implemented at a specific time. Qualitative reports of ammonia use in strength athletes suggest that this strategy is either implemented or not. To the authors knowledge no alternatives are employed.
We acknowledge the lack of control swill or appropriately blinded placebo is a limitation, however since alterations in legislation brought about due to a death associated with caffeine ingestion in 2018, obtaining institutional ethical approval for studies that involve ingestion of novel products has become increasingly difficult. The ethical committee felt that deception was inappropriate and would not approve the design under those conditions. Again, we acknowledge that a placebo swill would have strengthened the design and have included this as a limitation.
For a recreational trained population with little to no known experience with weight training, why did you only have 1 familiarization session? The learning curve for power based movement patterns is large.
In regard to minimising the learning effect, the researchers are in agreeance that allowing for a more robust familiarisation protocol to take place would be advantageous and note that Hopker (2009) and colleagues suggest at least 3 familiarisation sessions for sprinting. However, participant availability to an increased number of sessions was not viable under our current circumstances or time frames. It should also be noted that a study by Pekunlu (2015) indicated that a single familarization session with an extensive number of short cycling sprints can reduce variability. While Nibali (2015) and colleagues found vertical jump can be performed without the need for familiarisation.
Participants were also students on various sport and exercise courses that routinely exposed participants to movements and exercises at repetition ranges that were similar to the testing performed.
Hopker, J. G., Coleman, D. A., Wiles, J. D., & Galbraith, A. (2009). Familiarisation and reliability of sprint test indices during laboratory and field assessment. Journal of sports science & medicine, 8(4), 528–532. states
Pekünlü, E. (2015). A large number of short sprints is required to avoid practice based improvement effect during the assessment of cycling peak power. Isokinetics & Exercise Science, 23(2), 133–142.
Nibali, M. L. , Tombleson, T. , Brady, P. H. & Wagner, P. (2015). Influence of Familiarization and Competitive Level on the Reliability of Countermovement Vertical Jump Kinetic and Kinematic Variables. Journal of Strength and Conditioning Research, 29(10), 2827–2835. doi: 10.1519/JSC.0000000000000964.)
Did you determine test-retest reliability and variance for each participant to determine whether the experimental results were a meaningful change outside of individual variance?
We did not determine these metrics but agree that their calculation would allow for inference as to meaningful change. We did however use a ‘mean-best’ approach, whereby participants best two scores on each test under each condition were averaged. This approach accepts that participant variability in performance is inherent, but by excluding the lowest trial serves to minimise participant variance in performance measures to some extent.
Ln 144-156: Please discuss how the Wattbike was calibrated prior to each test set.
The following statement has been included in text ‘Wattbikes are factory calibrated so detail as to calibration is proprietary, although to ensure consistency between participants a factory reset was performed between participants.’ We accept that this is problematic, and the manufacturer does provide detail as to how to undertake a pseudo-calibration which assesses the level of agreement between cadence, target power output and airbrake resistance, but felt at the time that a factory reset was more appropriate and time efficient.
Why was a perceptual index not used (i.e. sRPE) to assess the perceived affects on performance with the use of the menthol swill?
Given the maximal nature of the tests performed we felt that the addition of a measure such as sRPE was not required. We appreciate that the inclusion of sRPE or similar would have served to confirm the maximal nature of the test, but would question when and how this would best be implemented. After each trial would seem excessive, and may risk committing a type 1 error due to ‘over-measurement’, and would not align with the statistical approach taken for assessing participant performance (mean of two best trials, per test). If administered at the end of the testing session or 30 minutes’ post-session, this may suggest that participants had either not performed maximally, or feel that they had performed maximally but recovered more quickly. The advantage of the latter is unclear, but we acknowledge this may be of interest in future.
Ln 225 – Please indent new paragraphs
Amended as requested
Ln 420 – Inconsistent journal formatting.
Amended as requested
Overall, the methodology of this study has some major flaws – the methods are lacking detail, no deception was used or blinding to the purpose of the study, no placebo swill, minimal familiarization of the power based protocols. The authors acknowledge the limitations in the discussion, which is good, but it is alarming that the methods were not probably vetted prior to the administration of this study.
We thank the reviewer for their appraisal of the manuscript and highlighting their methodological concerns. We acknowledge that the study is not perfect and indeed could be improved but have operated within the current institutional ethical guidelines in order to obtain approval to conduct the study. We have highlighted the methodological concerns further, in line with the reviewer’s recommendations.
Round 2
Reviewer 1 Report
Thank you for your responses to my comments. The additions to the paper and rebuttal where relevant are positive and satisfactory. I just have one comment re line 217 where the sentence end is missing some detail and needs amending.
Reviewer 2 Report
Thank you for responding to my suggestions in a clear and timely way.